# Changes in the analgesic mechanism of oxytocin can contribute to hyperalgesia in Parkinson's disease model rats

**Nayuka Usami** *, **Hiroharu Maegawa**, **Masayoshi Hayashi, Chiho Kudo, Hitoshi Niwa**

Department of Dental Anesthesiology, Osaka University Graduate School of Dentistry, Suita, Osaka, Japan

* nayuka.usami@gmail.com

## Abstract

Pain is a major non-motor symptom of Parkinson's disease (PD). Alterations in the descending pain inhibitory system (DPIS) have been reported to trigger hyperalgesia in PD patients. However, the underlying mechanisms remain unclear. In the current study, dopaminergic nigrostriatal lesions were induced in rats by injecting 6-hydroxydopamine (6-OHDA) into their medial forebrain bundle. The neural mechanisms underlying changes in nociception in the orofacial region of 6-OHDA-lesioned rats was examined by injecting formalin into the vibrissa pad. The 6-OHDA-lesioned rats were seen to exhibit increased frequency of face-rubbing and more c-Fos immunoreactive (c-Fos-IR) cells in the trigeminal spinal subnucleus caudalis (Vc), confirming hyperalgesia. Examination of the number of c-Fos-IR cells in the DPIS nuclei [including the midbrain ventrolateral periaqueductal gray, the locus coeruleus, the nucleus raphe magnus, and paraventricular nucleus (PVN)] showed that 6-OHDA-lesioned rats exhibited a significantly lower number of c-Fos-IR cells in the magnocellular division of the PVN (mPVN) after formalin injection compared to sham-operated rats. Moreover, the 6-OHDA-lesioned rats also exhibited significantly lower plasma oxytocin (OT) concentration and percentage of oxytocin-immunoreactive (OT-IR) neurons expressing c-Fos protein in the mPVN and dorsal parvocellular division of the PVN (dpPVN), which secrete the analgesic hormone OT upon activation by nociceptive stimuli, when compared to the sham-operated rats. The effect of OT on hyperalgesia in 6-OHDA-lesioned rats was examined by injecting formalin into the vibrissa pad after intracisternal administration of OT, and the findings showed a decrease in the frequency of face rubbing and the number of c-Fos-IR cells in the Vc. In conclusion, these findings confirm presence of hyperalgesia in PD rats, potentially due to suppression of the analgesic effects of OT originating from the PVN.

## Introduction

The classical motor symptoms of Parkinson's disease (PD), a neurodegenerative disorder that causes widespread loss of dopaminergic neurons in the nigrostriatal pathway, include resting tremors, bradykinesia (slow movement), muscle rigidity, and postural instability [1, 2]. However, 30%–85% of PD patients exhibit non-motor symptoms such as pain, hyperalgesia, and allodynia first, suggesting neurodegeneration of the noradrenergic and serotonergic nerves in addition to the dopaminergic nerves [3–14].

**Funding:** We received funding from JSPS KAKENHI (Grant number: 26463062). The funders involved in Conceptualization, Data curation, Formal analysis, Supervision, Validation, Visualization, and Writing – review & editing.

**Competing interests:** The authors have declared that no competing interests exist.

Previous evidence suggests that PD model rats induced by unilateral and bilateral injection of 6-hydroxydopamine (6-OHDA) into the medial forebrain bundle (MFB), striatum, or substantia nigra can result in allodynia and hyperalgesia after subcutaneous (SC) injection of formalin into the hind limbs and vibrissa pad [15–21]. However, the mechanism underlying this hyperalgesia remains unclear, with some studies suggesting that changes in the descending pain inhibitory system (DPIS) associated with reduced function of the nigrostriatal pathway may facilitate excitatory neurotransmission in the spinal dorsal horn and trigeminal spinal subnucleus caudalis (Vc) leading to hyperalgesia [3, 22–27]. When the limbic system is activated in response to painful stimuli, endogenous opioids disinhibit GABAergic neurons in the midbrain ventrolateral periaqueductal gray (vlPAG) and activate the locus coeruleus (LC) and the nucleus raphe magnus (NRM) which, in turn, leads to stimulation of the DPIS mediated by the noradrenergic and serotonin systems [17, 25, 28–30]. Evidence also suggests that the paraventricular nucleus (PVN) may function as a DPIS [22, 25, 31, 32]. We examined the relationship between PVN and hyperalgesia in PD as certain studies have reported a substantial decrease in the c-Fos-immunoreactive (-IR) count in the PVN after the SC injection of formalin into the vibrissa pad or the hind paws of PD model rats [19, 21].

Histologically, the PVN is composed of magnocellular and parvocellular neurons that play an important role in antinociception and stress response [33–35]. Acute/chronic nociceptive stimuli can activate the PVN and facilitate neuroendocrine responses such as stimulation of the hypothalamopituitary-adrenocortical (HPA) axis and the oxytocin (OT) and vasopressin (VP) systems which are involved in the modulation of nociceptive afferent pathways [33, 34, 36–39]. OT-containing neurosecretory cells in the magnocellular division of the PVN (mPVN) deliver OT to the axon terminals where it is stored in vesicles, exocytosed into the blood vessels from the posterior pituitary (PP), and then delivered to the target organs through the systemic circulation [33]. OT-containing neurosecretory cells in the dorsal parvocellular division of the PVN (dpPVN) project directly into the brainstem (including Vc) and spinal cord, and are associated with autonomic regulation and antinociception [22, 33–35, 40, 41].

Although several reports have suggested that alterations in the neural activity of the PVN, one of the DPIS, are associated with hyperalgesia in PD model rats, the mechanisms underlying hyperalgesia in PD model rats remain unclear, and no reports have investigated the association between hyperalgesia and OT originating from the PVN. The present study aimed to investigate the role of DPIS and involvement of OT in mitigating hyperalgesia in PD model rats with dopaminergic nigrostriatal lesions induced by 6-OHDA, and to understand the mechanisms underlying hyperalgesia in PD. We hypothesized that hyperalgesia in PD model rats is accompanied by altered nociceptive afferent pathways in the trigeminal region towing to the suppression of the analgesic effects of OT originating from the PVN and that OT administration may attenuate pain in these animals. Therefore, we developed unilateral PD model rats by administering 6-OHDA into the MFB and performed the formalin test into the vibrissa pad in rats as a chemical stimulation. We then examined immunohistochemical reactions in DPIS and OT neurosecretory cells of the PVN and changes in blood OT levels. We also administered OT via the intracisternal route to PD model rats to verify the behavioral and immunohistochemical responses to the SC injection of formalin into the vibrissa pad.

## Materials and methods

### Animals

A total of 95 male Wistar rats (Japan Lab Animals Co., Ltd., Osaka, Japan; weight: 150–200 g) were used in this study. The rats were housed under a 12 h dark/light cycle and were given access to food and water *ad libitum*. The study was approved by the Osaka University Graduate

School of Dentistry Animal Care and Use Committee, and all experimental procedures were performed in accordance with the National Institute of Health Guide for the Care and Use of Laboratory Animals. The present study builds on previous research demonstrating that hyperalgesia is induced in PD rats [18]. Only male rats were used in the present study because they were the only ones used in the previous study and it was necessary to conduct the experiment under the same conditions as it was conducted previously. It will certainly be necessary in the future to investigate whether the same results can be achieved in a similar study on hyperalgesia involving female rats as it was in male rats.

## Inducing 6-OHDA lesions

The procedures used to induce PD in the rats have been reported previously [18]. Briefly, the rats were first anesthetized using intraperitoneal (IP) administration of a saline solution containing midazolam (2.0 mg/kg, Sandoz, Tokyo, Japan), medetomidine (0.375 mg/kg, Zenoaq, Fukushima, Japan), and butorphanol (2.5 mg/kg, Meiji Seika Pharma, Tokyo, Japan). Thereafter, the head hair was shaved, the animals were immobilized in a stereotaxic apparatus (Narishige, Tokyo, Japan), and a small hole was created in the skull using a dental drill. Unilateral nigrostriatal lesions were created by injecting 6-OHDA (Sigma, St. Louis, MO, USA; 15μg in 5μl of sterile saline containing 0.01% ascorbic acid) into the left MFB. The stereotaxic coordinates of the lesions were 3.3 mm rostral to the interaural line, 1.4 mm left of the midline, and 6.8 and 6.5 mm ventral to the dural surface (2.5μl injected in each location). The 6-OHDA solution was administered through a cannula with a microinjection pump (Nihon Kohden, Tokyo, Japan) at a rate of 1 μl/min. The cannula was left in place for 5 min after completion of each injection before being slowly removed. The hole was filled with bone wax thereafter. The sham-operated rats received saline injections instead of the 6-OHDA solution.

## Rotational behavior test

Two weeks after injecting 6-OHDA or saline into the left MFB, the rats were placed in a cylindrical container (300 mm in diameter) and IP administration of methamphetamine (3 mg/kg, Dainippon Sumitomo Pharma Co., Ltd., Osaka, Japan) was performed to trigger rotational behavior [42]. The frequency of occurrence of the behavior was recorded using a video camera over a period of 1 hour after methamphetamine administration, and animals exhibiting ≧7 turns/minute were identified as 6-OHDA-lesioned rats exhibiting 6-OHDA lesions [17, 18, 43, 44]. The sham-operated rats were also injected with methamphetamine, although they did not exhibit any rotational behavior.

## Study protocol

The experimental outline is illustrated in Fig 1. 6-OHDA-lesioned rats were evaluated through the methamphetamine-induced rotational behavior test and immunoreactivity for tyrosine hydroxylase (TH), which are markers of dopaminergic neuron marker, to characterize the PD model in all experiments. The sham-operated rats were also injected with methamphetamine in all experiments. In the first experiment, three weeks after 6-OHDA or saline injection into the left MFB, an orofacial formalin test was performed to allow measurement of the response to chemical stimulation. The rats were randomly divided into three groups according to the SC injection stimulation: no injection, saline injection, and formalin injection. To elucidate whether 6-OHDA-lesioned rats experienced hyperalgesia after the SC injection of formalin, we performed behavioral studies and immunoreactivity analyses for c-Fos (a marker of activated neuronal nuclei) in the Vc. To investigate the role of DPIS, we performed immunoreactivity analyses for c-Fos in the vlPAG, PVN, LC, and NRM [25, 32, 34, 45]. The vlPAG was

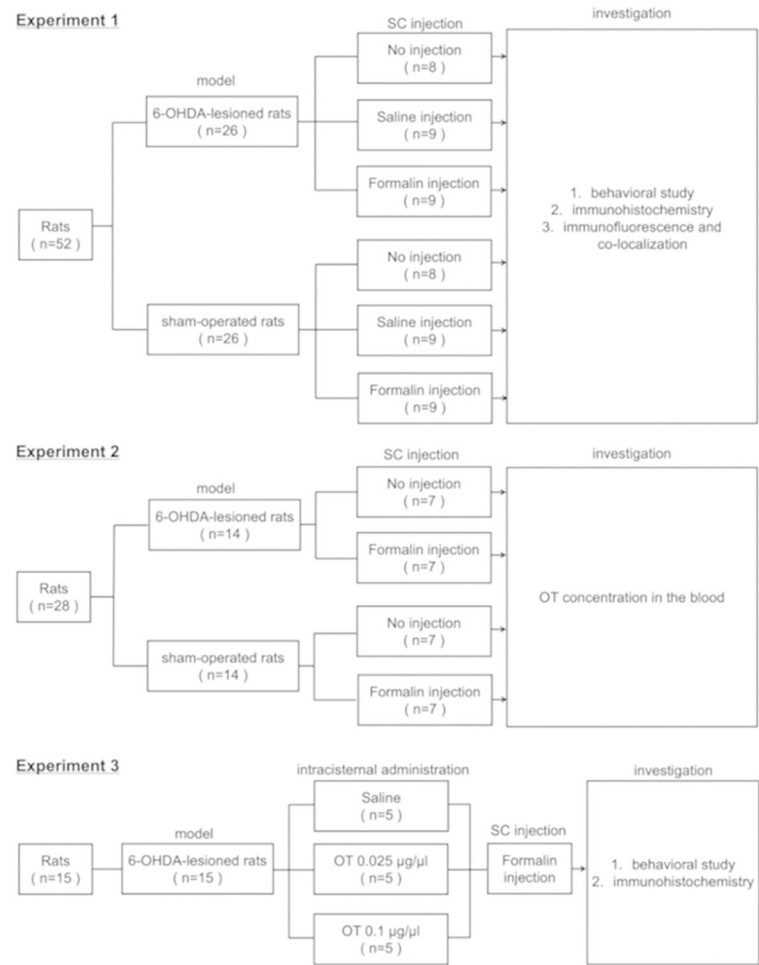

**Fig 1. Experimental outline of the study.** The study consists of three experiments. 6-OHDA: 6-hydroxydopamine, SC: subcutaneous, OT: oxytocin.

examined because of its involvement in pain modulation and opiate-induced antinociception [17, 46]. Subsequently, we performed immunofluorescent staining for c-Fos and OT to investigate the involvement of OT in the mitigation of hyperalgesia in 6-OHDA-lesioned rats. In the second experiment, 6-OHDA-lesioned rats and sham-operated rats were randomly divided into two groups, which are the no-injection and formalin-injection groups. Blood OT levels in rats were then assessed by collecting blood samples under no injection or 15 minutes after the SC injection of formalin. In the third experiment, 6-OHDA-lesioned rats were administered OT or saline via the intracisternal route, followed by the SC injection of formalin. Furthermore, we performed behavioral studies and immunoreactivity analyses for c-Fos in the Vc to investigate whether OT alleviates hyperalgesia in 6-OHDA-lesioned rats.

## Orofacial formalin test

The rats were first habituated individually in an observation chamber (25 × 25 × 25 cm and made of transparent Plexiglas) for at least 60 minutes. The saline injection group received a 50-μl SC injection of saline into the vibrissa pad on the same side as the 6-OHDA injection (henceforth referred to as the ipsilateral side) using a 26-gauge needle and minimal restraint.

The formalin group received a 50-μl SC injection of 4% formalin dissolved in saline in the vibrissa pad on the ipsilateral side using a 26-gauge needle with minimal restraint. Following injection, the rats were immediately placed back in the observation chamber and their behavior was recorded using a video camera for 45 minutes. The no injection group did not receive any injection.

The rats did not receive any food or water during the observation period, and the observation chamber was cleaned after each animal. The 45-minute recording period was divided into 9 blocks of 5 minutes each, and pain-related behavior was recorded by examining the frequency of face-rubbing in the direction of the injected area in the first (0–10 minutes after formalin injection) and second (10–45 minutes after formalin injection) phases [33, 47]. All recordings were made by one examiner who was blinded to the experimental procedures carried out.

## Immunohistochemistry

Two hours after SC injection of formalin or saline, the rats were anesthetized using an IP overdose injection of sodium pentobarbital and intracardial perfusion was performed using 150 ml of phosphate-buffered saline (PBS, 0.02 M, pH 7.4) followed by 500 ml of 4% (w/v) paraformaldehyde in PBS. Thereafter, the brain was removed, post-fixed at 4˚C for 24 hours, and then transferred to 30% sucrose in 0.01 M PBS at 4˚C for 48 hours. Serial coronal sections were made using a freezing microtome, and all sections were stored in PBS.

For TH immunostaining, the sections containing the striatum and substantia nigra pars compacta (40μm thickness) were incubated with 0.3% hydrogen peroxide in methanol for 20 minutes, and then rinsed in PBS 3 times for 5 minutes each. The sections were then blocked with 1% normal horse serum (NHS, Vector Labs, Burlingame, CA, USA) and 0.1% Triton X-100 in PBS for 30 minutes at room temperature (RT), before being incubated overnight at 4˚C with mouse anti-TH antibody (1:8000 dilution, T2928, Sigma-Aldrich, St. Louise, MO, USA), 1% NHS, and 0.1% Triton X-100 in PBS. The sections were then rinsed 3 times in PBS for 5 minutes each, incubated with biotinylated horse anti-mouse antibody (1:200 dilution, Vector Labs, Burlingame, CA, USA) for 1 hour at RT, and then rinsed 3 times in PBS again. The sections were then incubated with avidin-biotin-peroxidase complex (1:200 dilution, ABC Elite kit, Vector Labs) for 1 hour, treated with the DAB Substrate kit (Sigma-Aldrich), mounted on glass slides, air-dried, dehydrated, and cover-slipped.

For immunostaining of the c-Fos, the sections containing PVN (40μm thickness), vlPAG, NRM, LC, and Vc (60μm thickness) were incubated with 0.3% hydrogen peroxide in methanol for 20 minutes and then rinsed 3 times in PBS for 5 min each. Blocking was carried out using 1% normal goat serum (NGS, Vector Labs) and 0.1% Triton X-100 in PBS, after which the sections were incubated overnight at 4˚C with rabbit anti-cFos antibody (1:6400 dilution, #2250, Cell Signaling Technology, Danvers, MA, USA), 1% NHS, and 0.1% Triton X-100 in PBS. The sections were then rinsed 3 times in PBS, incubated with biotinylated goat anti-rabbit antibody (1:200 dilution, Vector Labs, Burlingame, CA, USA) for 1 hour at RT, and then rinsed 3 times in PBS again. Thereafter, the sections were incubated with avidin-biotin-peroxidase complex (1:200 dilution, ABC Elite kit, Vector Labs) for 1 hour, counter-stained with cresyl violet, and visualized with a DAB kit after being mounted on glass slides, air-dried, dehydrated, and cover-slipped.

For immunofluorescent staining of c-Fos and OT, the sections containing the PVN (40μm thickness) were blocked with 10% NGS and 0.3% Triton X-100 in PBS for 1 hour at RT, and then incubated overnight at 4˚C with rabbit anti-c-Fos antibody (1:1000 dilution, #2250, Cell Signaling Technology), mouse anti-OT antibody (1:1000 dilution, MAB5296, Millipore,

Burlington, MA, USA), 2% NGS, and 0.3% Triton X-100 in PBS. The sections were then rinsed 3 times in PBS and incubated with goat anti-rabbit Alexa Fluor 488 (1:200 dilution, Invitrogen, Carlsbad, CA, USA), goat anti-mouse Alexa Fluor 568 (1:200 dilution, Invitrogen), and 2% NGS in PBS for 2 hours at RT. After rinsing 3 times in PBS again, the sections were mounted on glass slides and cover-slipped. The remainder of the sections were used as negative controls for immunofluorescent staining. No positive products were observed upon omission of the primary antibodies.

## Cell counting

The number of c-Fos immunoreactive (c-Fos-IR) cells in the PVN, vlPAG, and LC on the ipsilateral and contralateral (to 6-OHDA injection into the MFB) sides were counted, and further grouping by mPVN, mpPVN, and dpPVN was also done as the neurons in these divisions vary in terms of the physiological signaling systems they control [22, 33–35, 40, 41].

The location coordinated of the PVN, vlPAG, NRM, LC, and Vc were determined using the Paxinos and Watson atlas [48]. Images were captured using a light microscope (BX51, OLYMPUS, Tokyo, Japan) or a confocal laser microscope (Zeiss LSM700) with a 10x objective lens. The c-Fos-IR cells were identified using a black reaction product that was confined to the cell nucleus, and all cells that were darker than the surrounding background were counted. To quantify the proportion of OT neurons activated by SC injection of formalin into the left vibrissa pad, the percentage of OT-immunoreactive (-IR) neurons expressing c-Fos protein in the PVN was assessed by manually counting the OT-IR neurons and OT-IR neurons co-localized with c-Fos protein and then using the following formula:

$$\left( \frac{\text{number of OT−IR neurons co−localized with c−Fos protein}}{\text{number of OT−IR neurons}} \right) \times 100 \ (\%)$$

Two sections in the PVN; three sections in the vlPAG, LC, and NRM; and seven sections in the Vc per rat were selected in accordance with previous literature [33, 34, 49–51]. The number of cells was counted by one examiner who was blinded to the experimental procedures carried out.

## Measurement of oxytocin concentration in the blood

As the blood OT concentration is influenced by the circadian rhythm, all experiments were carried out between 9am and 3pm three weeks after the injection of 6-OHDA or saline into the left MFB [52].

The rats in the formalin injection group were anesthetized using IP sodium pentobarbital (50 mg/kg), followed by formalin injection into the vibrissa pad and 15 minutes later, 3.0 ml of blood was collected from the heart. The rats in the no injection group were anesthetized using IP sodium pentobarbital and 3.0 ml of blood was collected from the heart under no injection. The blood was then placed in chilled Eppendorf tubes containing ethylenediaminetetraacetic acid (1.0 mg/ml), and plasma samples were obtained by centrifugation (KITMAN-18, TOMY DIGITAL BIOLOGY Co. Ltd., Tokyo, Japan) at 4°C and 1400×g for 15 minutes) [33, 53]. The plasma samples were stored in aliquots at -70°C, and OT concentration in the plasma was analyzed using an enzyme immunoassay (EIA) kit (Enzo Life Science Inc., NY, USA). Samples were thawed only once just before the assay and diluted 4 times to match the measurement range (15.6–1000 pg/ml). Luminescence counts were measured using the xMark™ Microplate Absorbance Spectrophotometer (Bio-Rad Laboratories Inc., CA, USA).

### Intracisternal administration of oxytocin before formalin test

The cisterna magna of the 6-OHDA-lesioned rats were cannulated using the modified method described previously by Sarna et al., [54–56]. After rotational behavior tests, the rats were anesthetized using a saline solution containing midazolam (2.0 mg/kg), medetomidine (0.375 mg/kg), and butorphanol (2.5 mg/kg). The head hair was then shaved, the skull was cleaned and dried, and a microdrill was used to create a hole 2 mm to the left of the midline and 5 mm caudal to the lambda to allow catheter placement. Two additional holes were created symmetrically a few millimeters anterior and lateral to the first hole for placement of micro screws. The intracisternal catheters, which were silicon tubes (5cm in length, 0.5 mm in diameter; eastsidemed Inc., Tokyo, Japan) filled with sterilized saline, were inserted into the subarachnoid space through the hole and the caudal end was passed gently into the cisterna magna. The length of the silicon tube from the hole to the caudal tip was 6 mm. After positioning, the catheter was flushed with normal saline and a clear fluid was seen to move up through the tube. Dental acrylic (PROVISTA®, Sun Medical Company Ltd., Shiga, Japan) was applied around the catheter and the screws to hold them in place and the incision was closed and sutured. The open end of the catheter was then plugged with a removable plastic fiber. Only rats that did not exhibit post-surgical motor dysfunction were used for the subsequent experiment. Two days after placement, the catheter was flushed with 10μl of saline for transparency and, seven days after cannulation surgery, the formalin test was carried out after intracisternal administration of OT (0.25 or 1.0 μg in 10 μl of saline; H-2510, Bachem Holding AG, Bubendorf, Switzerland) or saline [56]. After each administration of 10μl of the OT solution, the catheter was filled with 10μl of saline for flushing. a 50-μl SC injection of 4% formalin was administered into the left vibrissa pad after 1 minute using a 26-gauge needle, and the behavior of the rats was recorded for 45 minutes using a video camera. Two hours after injection of formalin, intracardial perfusion was carried out and the brains were post-fixed and transferred to 30% sucrose in the same way as mentioned above. For immunostaining of the c-Fos using DAB, the sections were incubated and the number of c-Fos-IR cells in the Vc were counted as described above.

### Statistical analyses

Data were presented as mean ± standard error (SEM). Unpaired $t$-test was used to compare the difference between 6-OHDA-lesioned rats and sham-operated rats regarding behavioral and immunorhistochemical data after formalin test. To examine the analgesic effect of intracisternal administration of OT on hyperalgesia in 6-OHDA-lesioned rats, the behavioral and immunorhistochemical data were analyzed using one-way analysis of variance (ANOVA) followed by Bonferroni's test for multiple comparisons. Other results were analyzed using two-way ANOVA followed by Bonferroni's test for multiple comparisons. All statistical analyses were carried out using the statistical software, SPSS (IBM, Statistics ver. 24, IL, USA). A $p$-value $< 0.05$ was considered statistically significant.

## Results

### TH immunoreactivity

TH immunohistochemical staining was performed in all rats in the current study, and the sham-operated rats were seen to exhibit high TH immunoreactivity in the striatum and substantia nigra on both sides [Fig 2A and 2C)]. In contrast, TH immunoreactivity in the striatum and substantia nigra of the 6-OHDA-lesioned rats was markedly decreased on the ipsilateral side [left; Fig 2B and 2D] and remained the same as the sham-operated rats on the contralateral side.

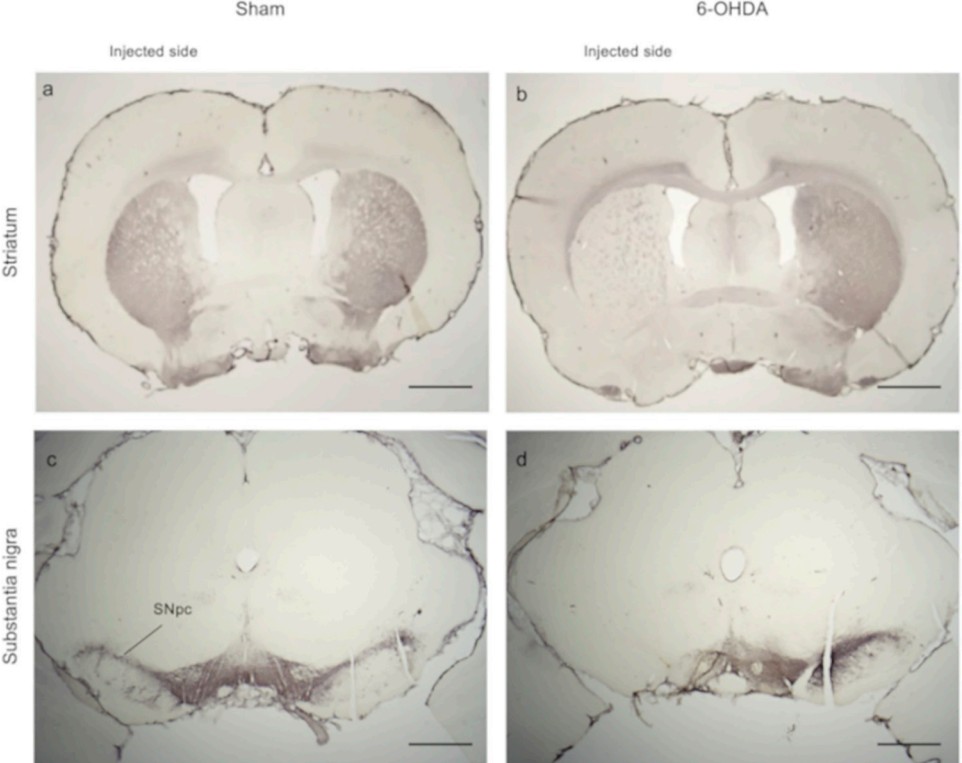

**Fig 2. Effect of 6-hydroxydopamine (6-OHDA) injection into the left medial forebrain bundle (MFB).** Tyrosine hydroxylase (TH) immunostaining of the striatum and substantia nigra pars compacta (SNpc) in the sham-operated group and 6-OHDA-lesioned group. No changes in TH immunoreactivity were observed on the saline injected side in the (a) striatum and the (c) substantia nigra, while decreased TH immunoreactivity was observed in the (b) striatum and (d) substantia nigra on the 6-OHDA injected side. Scale bars: (a, b) 1 mm; (c, d) 500 μm.

## Effect of formalin injection into the vibrissa pad on face-rubbing behavior and c-Fos expression in the Vc

The SC injection of formalin into the vibrissa pad induced face rubbings with face-wash strokes directed to the perinasal area with the ipsilateral forepaw, and sometimes with the hind paw, were observed. It has been reported that as formalin-induced face rubbing is sometimes bilateral and resemble normal face washing, a control group without formalin injection (i.e. saline injection group) should be used to quantify spontaneous, symmetrical face washing and provide a 'pain-free' baseline when performing orofacial formalin test [57, 58]. As in previous reports [15, 57, 58], formalin injection resulted in a biphasic increase in the number of face rubbing, but saline injection did not result in this response [Fig 3A(a, c), S1 File]. The increase of face rubbings during the first 10 minutes after SC injection of formalin or saline into the left vibrissa pad was defined as the first phase and that from 10 to 45 minutes as the second phase, and the mean frequency of face-rubbing in the first and second phases was measured. No significant difference in the frequency of face-rubbing was observed between the 6-OHDA-lesioned and sham-operated rats in the saline injection group [first phase: 19.3 ± 3.7 vs. 15.1 ± 4.3; second phase: 50.7 ± 8.1 vs. 31.8 ± 7.9; Fig 3A(b), S1 File]. In contrast, 6-OHDA-lesioned rats exhibited significantly higher frequency of face-rubbing than the sham-operated rats in the formalin injection group [first phase: 129.9 ± 22.6 vs. 71.8 ±14.2, p-value < 0.05; second phase: 703.2 ± 87.5 vs. 340.7 ± 60.6, p-value < 0.01; Fig 3A(d), S1 File].

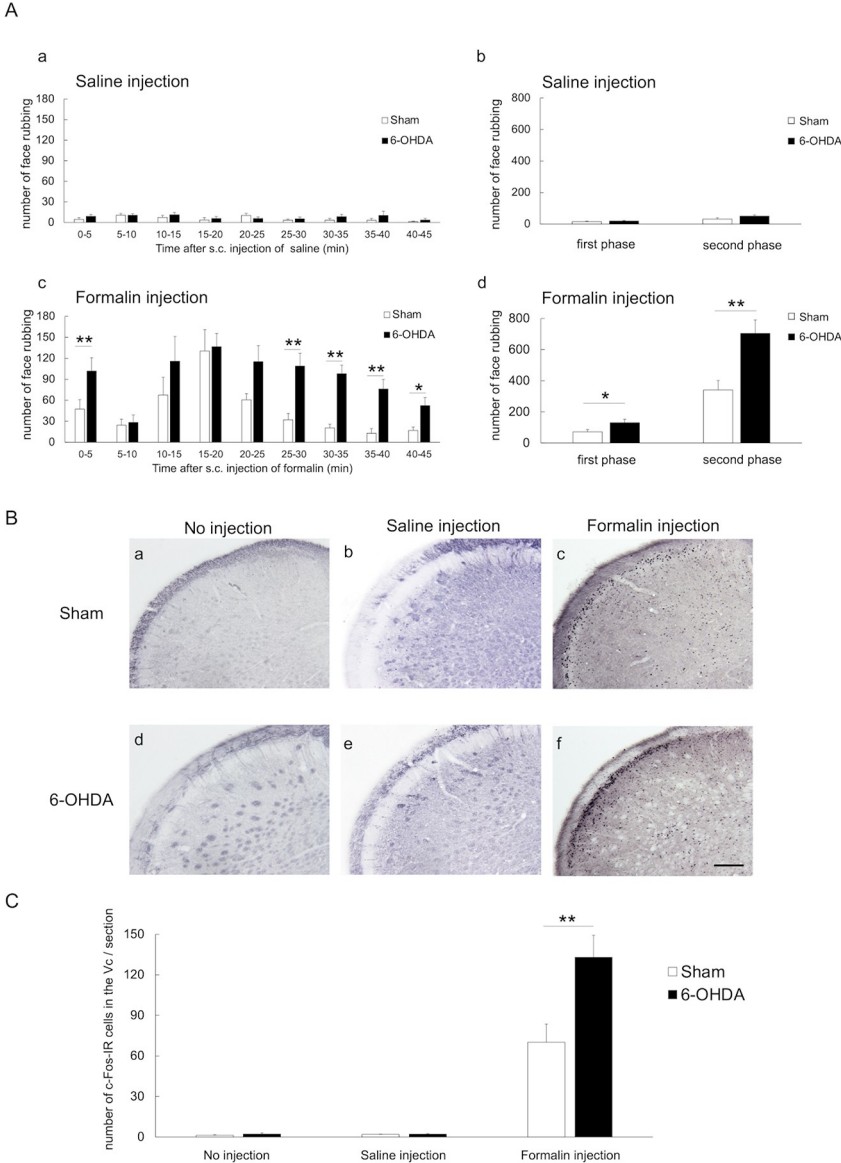

**Fig 3. Response to formalin injection in the vibrissa pad.** (A) The frequency of face-rubbing behavior after SC injection of saline or formalin into the left vibrissa pad (a, c) per 5 minutes, (b, d) in the first (0–10 minute) and second (10–45 minutes) phases: (a, b) the 6-OHDA-lesioned and sham-operated rats did not exhibit significant differences in the frequency of face-rubbing after SC administration of saline; (c) SC injection of formalin induced a two-phase increase in face rubbing behavior in both 6-OHDA-lesioned rats and sham-operated rats; (d) the 6-OHDA-lesioned rats exhibited significant increases in the frequency of face-rubbing in both phases compared to the sham-operated rats after SC injection of formalin. SC: subcutaneous; ** p-value < 0.01; * p-value < 0.05, unpaired *t*-test. (B) Photomicrographs of immunohistochemical staining of c-Fos in the trigeminal spinal subnucleus caudalis (Vc) in the (a, d) no injection; (b, e) saline injection; and (c, f) formalin injection groups. The photomicrographs show sections from the area 1440μm caudal to the obex and on the ipsilateral side. The findings showed c-Fos-immunoreactive (c-Fos-IR) cells primarily in the superficial layers of the Vc. Scale bar: 200 μm. (C) In the formalin group, the number of c-Fos-IR cells in the Vc were significantly increased in the 6-OHDA-lesioned rats compared to the sham-operated rats. However, the 6-OHDA-lesioned and sham-operated rats in the no injection and saline injection groups did not exhibit significant differences in the number of c-Fos-IR cells in the Vc. ** *p-value* < 0.01, unpaired *t*-test.

All three treatment groups (i.e., no injection, saline injection, and formalin injection) underwent immunohistochemical staining for c-Fos in the Vc (Fig 3B), and the number of c-Fos-IR cells in the superficial layers of the Vc was counted [55, 59–61]. Comparison of the 6-OHDA-lesioned and sham-operated rats showed no significant differences in the number of c-Fos-IR cells in the Vc in the no injection and saline injection groups (no injection: $1.8 \pm 0.7$ vs. $1.0 \pm 0.4$; saline injection: $1.8 \pm 0.3$ vs. $1.8 \pm 0.3$; Fig 3C, S1 File). In the formalin injection group, 6-OHDA-lesioned rats exhibited a significant increase in the number of c-Fos IR cells in the Vc compared to the sham-operated rats ($100.4 \pm 14.0$ vs. $55.1 \pm 10.7$, *p-value* $< 0.05$, Fig 3C, S1 File). SC injections of saline were used to investigate whether hyperalgesia in the PD model rats occurred due to chemical stimulation with formalin or mechanical stimulation with a needle. The findings confirmed that SC injection of saline did not cause hyperalgesia in PD model rats, and further investigation of the mechanism of hyperalgesia was then carried out by comparing the no injection and formalin injection groups.

## c-Fos expression in the DPIS

The number of c-Fos-IR cells in the DPIS nuclei (i.e., vlPAG, LC, NRM, and PVN) of the 6-OHDA-lesioned and sham-operated rats receiving formalin injections were counted (Fig 4A and 4B). No significant differences were observed in the DPIS except the PVN where a significant decrease in the number of c-Fos-IR cells was observed on the ipsilateral side of the 6-OHDA-lesioned rats compared to sham-operated rats (ipsilateral vlPAG: $22.8 \pm 3.3$ vs. $22.7 \pm 3.5$; contralateral vlPAG: $26.3 \pm 3.5$ vs. $19.3 \pm 2.9$; ipsilateral LC: $18.9 \pm 3.0$ vs. $24.4 \pm 5.2$; contralateral LC: $23.3 \pm 4.9$ vs. $21.6 \pm 4.7$; NRM: $14.1 \pm 2.7$ vs. $15.1 \pm 3.1$; ipsilateral PVN: $67.3 \pm 7.7$ vs. $109.5 \pm 17.0$, *p-value* $< 0.05$; contralateral PVN: $102.8 \pm 10.5$ vs. $95.7 \pm 16.2$; Fig 4B, S1 File). The PVN response to formalin injection was further investigated by counting the number of c-Fos-IR cells in the PVN of 6-OHDA-lesioned and sham-operated rats in the no injection and formalin injection groups (Fig 4C). In both 6-OHDA-lesioned and sham-operated rats, the number of c-Fos-IR cells in both ipsilateral and contralateral PVN was significantly increased in the formalin injection group compared to those in the no injection group [sham-operated ipsilateral PVN: $109.5 \pm 17.0$ vs. $31.4 \pm 8.3$, *p-value* $< 0.01$; 6-OHDA-lesioned ipsilateral PVN: $68.9 \pm 8.7$ vs. $21.6 \pm 4.8$, *p-value* $< 0.01$; sham-operated contralateral PVN: $91.3 \pm 14.6$ vs. $35.4 \pm 8.2$, *p-value* $< 0.01$, 6-OHDA-lesioned contralateral PVN: $102.8 \pm 10.5$ vs. $17.6 \pm 2.9$, *p-value* $< 0.01$, Fig 4C(a, b), S1 File]. In the formalin injection group, 6-OHDA-lesioned rats exhibited significant decreases in the number of c-Fos-IR cells in the ipsilateral PVN when compared to the sham-operated rats [$68.9 \pm 8.7$ vs. $109.5 \pm 17.0$, *p-value* $< 0.05$, Fig 4C(a), S1 File].

PVN neurons control three physiological signaling systems that differ by division, with neurons in the mPVN division expressing the OT and VP; neurons in the dpPVN division projecting to the brainstem and spinal cord (including Vc) and being associated with autonomic regulation and antinociception; and neurons in the mpPVN expressing CRH and regulating the HPA [22, 33–35, 41]. The PVN response to formalin injection was further investigated by counting the number of c-Fos-IR cells in the mPVN, dpPVN, and mpPVN of 6-OHDA-lesioned and sham-operated rats in the no injection and formalin injection groups (Fig 5A). In the formalin injection group, the 6-OHDA-lesioned rats exhibited significant decreases in the number of c-Fos-IR cells in the ipsilateral mPVN when compared to the sham-operated rats [$18.3 \pm 4.9$ vs. $38.4 \pm 9.3$, *p-value* $< 0.05$, Fig 5B(a), S1 File]. Moreover, a significant increase in the number of c-Fos-IR cells in the mPVN, dpPVN and mpPVN on the ipsilateral side of the sham-operated group and contralateral side of the 6-OHDA-lesioned group was observed in the formalin injection group compared to the no injection group [sham-operated ipsilateral

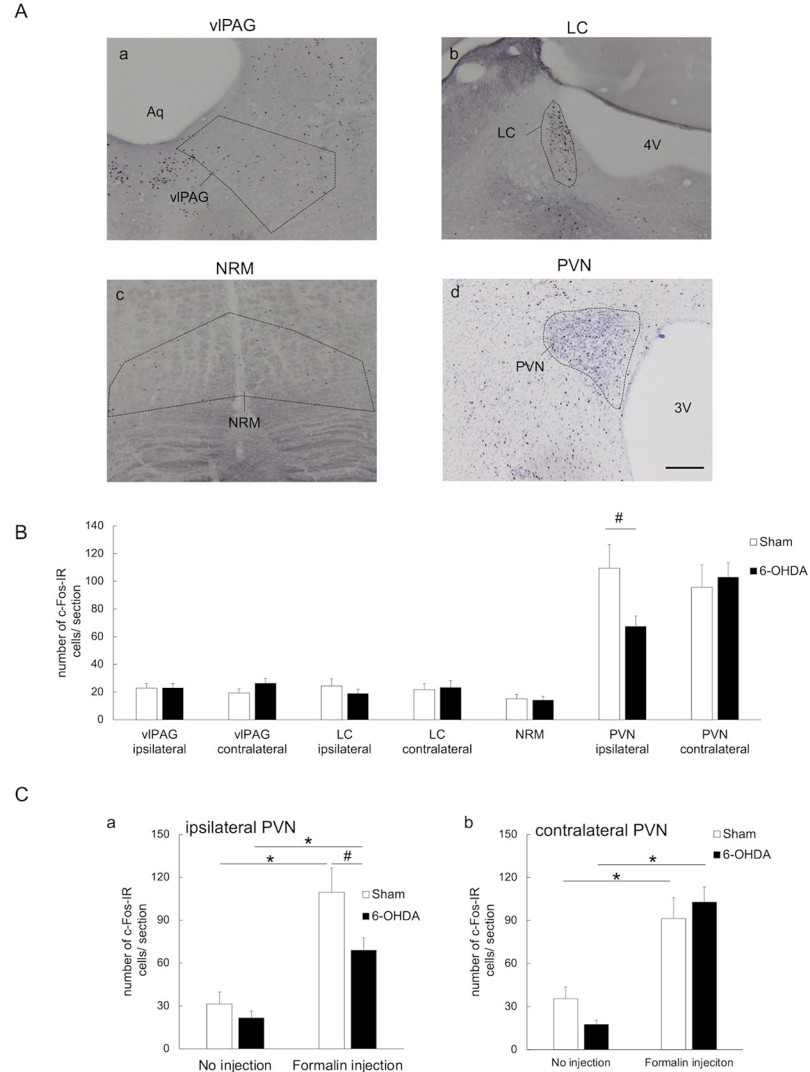

**Fig 4. Immunohistochemical evaluation of PAG, LC, NRM, and PVN activity after formalin injection.** (A) Photomicrographs of c-Fos-immunoreactive (c-Fos-IR) cells in (a) the ventrolateral midbrain periaqueductal gray (vlPAG), (b) the locus coeruleus (LC), (c) the nucleus raphe magnus (NRM), and (d) the paraventricular nucleus (PVN) of sham-operated rats receiving formalin injections. DAB was used to visualize c-Fos immunoreactivity. The sections of PVN were visualized using counter-staining with cresyl violet and DAB. Aq; midbrain aqueduct, 3V: third ventricle, 4V: fourth ventricle. Scale bars: 200 μm. (B) The number of c-Fos-immunoreactive (c-Fos-IR) cells in the vlPAG, LC, NRM, and PVN. The vlPAG was examined as a representative of the PAG which is known to be involved in pain modulation and opiate-induced antinociception. In the formalin injection group, the number of c-Fos-IR cells in the vlPAG, LC, and NRM of 6-OHDA-lesioned rats did not significantly differ from that of the sham-operated rats. However, significant decreases in the number of c-Fos-IR cells were observed in the PVN on the ipsilateral side. #$p$-$value < 0.05$, unpaired $t$-test. (C) In both 6-OHDA-lesioned and sham-operated rats, the number of c-Fos-IR cells in both ipsilateral and contralateral PVN was significantly increased in the formalin injection group compared to those in the no injection group. In the formalin injection group, 6-OHDA-lesioned rats exhibited significant decreases in the number of c-Fos-IR cells in the ipsilateral PVN when compared to the sham-operated rats. *$p$-$value < 0.01$ (vs. no injection group); #$p$-$value < 0.05$ (vs. sham-operated rats in formalin injection group), two-way ANOVA followed by Bonferroni's test for multiple comparisons.

mPVN: $38.4 \pm 9.3$ vs. $5.3 \pm 1.7$, $p$-$value < 0.01$; sham-operated ipsilateral dpPVN: $14.1 \pm 2.0$ vs. $3.8 \pm 1.5$, $p$-$value < 0.01$; sham-operated ipsilateral mpPVN: $56.9 \pm 8.6$ vs. $22.3 \pm 5.5$, $p$-$value < 0.01$; 6-OHDA-lesioned ipsilateral mPVN- $30.2 \pm 7.5$ vs. $4.2 \pm 1.1$, $p$-$value < 0.01$; 6-OHDA-lesioned ipsilateral dpPVN: $12.8 \pm 3.4$ vs. $2.2 \pm 0.4$, $p$-$value < 0.01$; 6-OHDA-

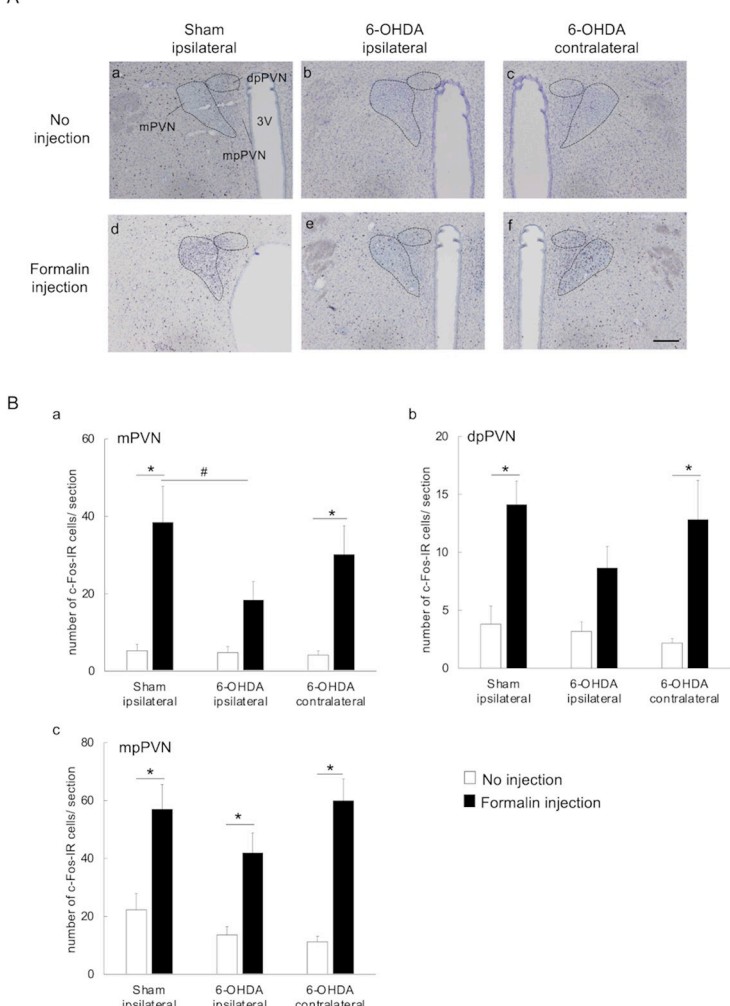

**Fig 5. c-Fos expression in each division of the paraventricular nucleus (PVN).** (A) Photomicrographs of c-Fos-IR cells in the PVN: sections from the ipsilateral side of sham-operated rats and the ipsilateral and contralateral sides of 6-OHDA-lesioned rats in the no injection and formalin injection groups were compared. The c-Fos-IR cells were visualized using counter-staining with cresyl violet and DAB. PVN: paraventricular nucleus; c-Fos-IR: c-Fos-immunoreactive; 3V: third ventricle; scale bar: 200 μm. (B) The number of c-Fos-IR cells in the mPVN, dpPVN, and mpPVN: (a) In the formalin injection group, the number of c-Fos-IR cells in the ipsilateral mPVN was significantly smaller in the 6-OHDA-lesioned rats compared to sham-operated rats; (a-c) the number of c-Fos-IR cells in the ipsilateral mPVN, dpPVN, and mpPVN were significantly higher in sham-operated rats in the formalin injection group compared to the no injection group; (a-c) the number of c-Fos-IR cells in the contralateral mPVN, dpPVN, and mpPVN were significantly higher in 6-OHDA-lesioned rats in the formalin injection group compared to the no injection group; (a, b) the number of c-Fos-IR cells in the ipsilateral mPVN and dpPVN did not significantly differ between 6-OHDA-lesioned rats in the formalin injection group compared to the no injection group. mPVN: magnocellular part of PVN; dpPVN: dorsal parvocellular part of PVN; mpPVN: medial parvocellular part of PVN; c-Fos-IR: c-Fos-immunoreactive; *$p$-value $< 0.01$ (vs. no injection group); #$p$-value $< 0.05$ (vs. ipsilateral side of sham-operated rats in formalin injection group), two-way ANOVA followed by Bonferroni's test for multiple comparisons.

lesioned ipsilateral mpPVN- 59.8 ± 7.6 vs. 11.2 ± 1.9, *p-value* < 0.01; Fig 5B(a-c), S1 File]. The number of c-Fos-IR cells in the ipsilateral mpPVN was significantly increased in 6-OHDA-lesioned rats in the formalin injection group compared to those in the no injection group [41.9 ± 6.9 vs. 13.6 ± 2.8, *p-value* < 0.01, Fig 5B(c), S1 File]; however, no such differences were observed in the mPVN and dpPVN [mPVN: 18.3 ± 4.9 vs. 4.8 ± 1.5, dpPVN: 8.6 ± 1.8 vs. 3.2 ± 0.8, Fig 5B(a, b), S1 File].

## Co-expression of c-Fos and OT in the PVN

Comparison of 6-OHDA-lesioned rats in the formalin injection and no injection groups showed no significant increases in the number of c-Fos-IR cells in the ipsilateral mPVN and dpPVN, suggesting altered function. Further examination showed the presence of OT-IR and c-Fos-IR cells in the mPVN and dpPVN [OT-IR: Fig 6A(a-d), c-Fos-IR: Fig 6A(e-h), merged images: Fig 6A(i-l); highly magnified images: Fig 6B]. To quantify the proportion of OT neurons activated by the SC injection of formalin into the left vibrissa pad, the percentage of OT-IR neurons expressing the c-Fos protein in the PVN was assessed. In the formalin injection group, significant decreases in the percentage of OT-IR neurons co-localized with c-Fos protein were observed in the ipsilateral mPVN and dpPVN of 6-OHDA-lesioned rats compared to sham-operated rats [mPVN: 10.4 ± 2.6 vs. 24.0 ± 3.9, $p$-value < 0.01; dpPVN: 30.4 ± 5.4 vs. 47.9 ± 1.6, $p$-value < 0.01; Fig 6C(a), S1 File]. The percentage OT-IR neurons co-localized with c-Fos protein in the mPVN and dpPVN was significantly higher in the formalin injection group compared to the no injection group [sham-operated ipsilateral mPVN: 24.0 ± 3.9 vs. 1.3 ± 0.3, $p$-value < 0.01, Fig 6C(a); sham-operated ipsilateral dpPVN: 47.9 ± 1.6 vs. 8.2 ± 3.2, $p$-value < 0.01, Fig 6C(b); 6-OHDA-lesioned ipsilateral mPVN: 10.4 ± 2.6 vs. 2.1 ± 0.5, $p$-value < 0.05, Fig 6C(a); 6-OHDA-lesioned ipsilateral dpPVN: 30.4 ± 5.4 vs. 3.8 ± 2.4, $p$-value < 0.01, Fig 6C(b); 6-OHDA-lesioned contralateral mPVN: 17.5 ± 2.8 vs. 1.3 ± 0.3, $p$-value < 0.01, Fig 6C(a); 6-OHDA-lesioned contralateral dpPVN: 39.3 ± 5.9 vs. 4.5 ± 1.6, $p$-value < 0.01, Fig 6C(b), S1 File].

## Effect of SC formalin injection on plasma OT

Plasma OT concentration in 6-OHDA-lesioned rats and sham-operated rats under no injection or 15 minutes after SC injection of formalin were assessed. The sham-operated rats in the formalin injection group exhibited higher plasma OT concentration than those in the no injection group (1701.0 ± 220.8 vs. 862.6 ± 124.4, $p$-value < 0.01, Fig 7, S1 File), although no such differences by injection status were observed in the 6-OHDA-lesioned rats (1031.0 ± 152.0 vs. 1218.4 ± 209.9, Fig 7). Moreover, the plasma concentration of OT was lower in 6-OHDA-lesioned rats in the formalin injection group compared to sham-operated rats in the same group (1031.0 ± 152.0 vs. 1701.0 ± 220.8, $p$-value < 0.05, Fig 7, S1 File).

## Effect of intracisternal administration of oxytocin

The orofacial formalin test was performed after intracisternal administration of OT (0.025 or 0.1 μg/μl) or saline in 6-OHDA-lesioned rats. In the formalin injection group, intracisternal administration of OT (0.025 or 0.1 μg/μl) did not affect the frequency of face-rubbing in the first phase (0.025 μg/μl OT: 87.8 ± 20.8; 0.1 μg/μl OT: 132.6 ± 95.3; saline: 160.6 ± 11.0; Fig 8A, S1 File). Administration of 0.1 μg/μl OT was seen to decrease the frequency of face-rubbing in the second phase to a greater extent than saline (295.4 ± 107.0 vs. 914.6 ± 147.0, $p$-value < 0.05; Fig 8A, S1 File), although no such differences were observed upon comparing intracisternal administration of 0.025 μg/μl OT and saline (445.4 ± 156.2 vs. 914.6 ± 147.0, Fig 8A, S1 File). Immunohistochemical staining for c-Fos in the Vc showed presence of c-Fos-IR cells in the superficial layers [Fig 8B (a-c)], with the number of cells being significantly lower in rats receiving 0.1 μg/μl OT compared to those receiving saline (15.8 ± 5.5 vs. 78.1 ± 4.1, $p$-value< 0.05, Fig 8C, S1 File). However, no such differences in the number of c-Fos-IR cells in the Vc were observed upon comparing rats receiving 0.025 μg/μl OT and those receiving saline (52.4 ± 20.8 vs. 78.1 ± 4.1, Fig 8C, S1 File).

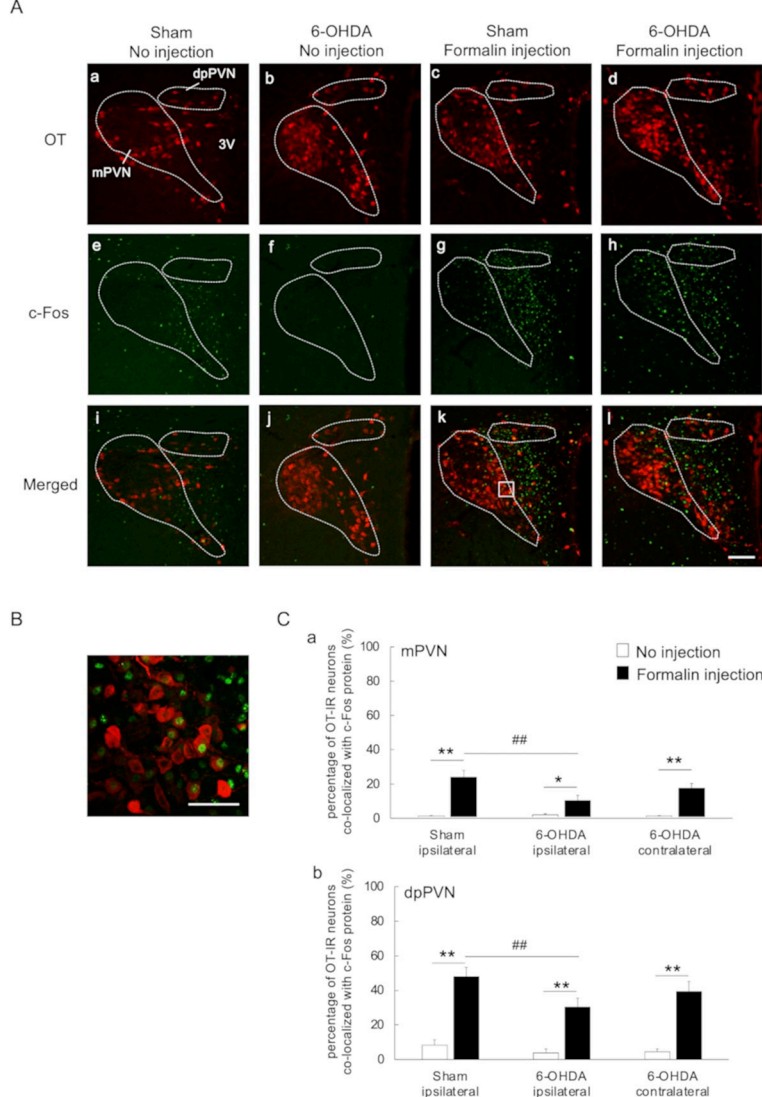

**Fig 6. Immunofluorescent labeling of OT and c-Fos in the paraventricular nucleus (PVN).** (A) Photomicrographs of oxytocin immunoreactive (OT-IR) cells (a-d), c-Fos immunoreactive (c-Fos-IR) cells (e-h), and merged images of the PVN (i-l). Sections from sham-operated rats in the no injection (a, e, i) and formalin injection (b, f, j) groups and 6-OHDA-lesioned rats in the no injection (c, g, k) and formalin injection (d, h, l) groups were examined. Scale bar: 100 μm. (B) Merged view obtained using high-power fluorescence microscopy (boxed region in Ak). Scale bar: 50 μm. (C) Changes in the percentage of oxytocin immunoreactive (OT-IR) neurons co-localized with c-Fos protein in the mPVN and dpPVN. The c-Fos-immunoreactive (c-Fos-IR) cells, OT-IR cells, and OT-IR cells co-localized with c-Fos protein in the mPVN and dpPVN of sham-operated and 6-OHDA-lesioned rats in the no injection and formalin injection groups were counted manually to allow calculation of the percentage of OT-IR neurons co-localized with c-Fos protein. 6-OHDA-lesioned rats in the formalin injection group exhibited a significantly smaller percentage of OT-IR cells co-localized with c-Fos protein in the ipsilateral mPVN (a) and dpPVN (b) when compared to sham-operated rats in the same group. Moreover, the sham-operated and 6-OHDA-lesioned rats in the formalin injection group exhibited an increased percentage of OT-IR neurons co-localized with c-Fos protein in the ipsilateral mPVN and dpPVN when compared to the no injection group (a, b). mPVN: magnocellular part of paraventricular nucleus; dpPVN: dorsal parvocellular part of paraventricular nucleus; $**p\text{-}value < 0.01$; $* p\text{-}value < 0.05$ (vs. no injection group); $^{\#}p\text{-}value < 0.01$ (vs. ipsilateral side of sham-operated rats in formalin injection group), two-way ANOVA followed by Bonferroni's test for multiple comparisons.

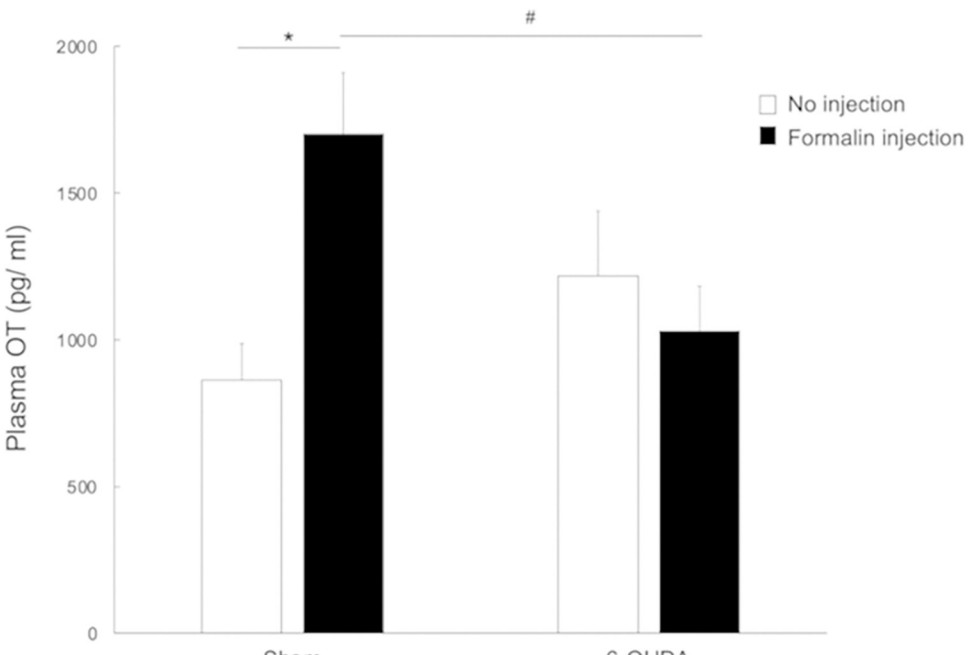

**Fig 7. Changes in the concentration of plasma oxytocin (OT).** Comparison of sham-operated rats by injection status showed markedly increased plasma OT concentration in the formalin injection group compared to the no injection group. In contrast, no such differences in plasma OT concentration by injection status were observed in the 6-OHDA-lesioned rats. Finally, the plasma concentration of OT was lower in 6-OHDA-lesioned rats in the formalin injection group compared to sham-operated rats in the same group. $*p$-value $< 0.01$ (vs. sham-operated rats in no injection group); $\#p$-value $< 0.05$ (vs. sham-operated rats in formalin injection group), two-way ANOVA followed by Bonferroni's test for multiple comparisons.

## Discussion

The findings of the current study showed that unilateral nigrostriatal lesions, created by injecting 6-OHDA into the MFB of rats, were associated with increased face-rubbing and c-Fos expression in the Vc after SC injection of formalin into the vibrissa pad. This was in agreement with previous studies that showed that rats with unilateral PD exhibited a hyperalgesic response to orofacial chemical stimulation [17, 18, 20, 21, 57, 62].

The mechanisms underlying this hyperalgesia were examined by measuring the c-Fos expression in brain regions related to the DPIS, and the findings showed no differences in the number of c-Fos-IR cells in the vlPAG, LC, and NRM between 6-OHDA-lesioned and sham-operated rats receiving formalin injections. However, in the formalin injection group, 6-OHDA-lesioned rats exhibited a significant decrease in the number of c-Fos-IR cells in the ipsilateral PVN when compared to the sham-operated rats, and this finding was also in agreement with previous studies [19, 21]. Further examination showed that the number of c-Fos-IR cells in the ipsilateral mPVN and dpPVN of 6-OHDA-lesioned rats did not significantly differ by injection status (i.e., formalin injection vs no injection groups). The PVN has been shown to increase the number of c-Fos-IR cells ipsilaterally and contralaterally after unilateral SC injection of formalin [34]. Moreover, the mPVN and dpPVN divisions contain a large number of neurosecretory cells that release OT, a neuropeptide known to exhibit analgesic properties [22, 33–35, 41]. Further examination of potential altered neural activity in these neurosecretory cells showed that the percentage of OT neurons co-localized with c-Fos protein in the ipsilateral mPVN and dpPVN was significantly smaller in 6-OHDA-lesioned rats compared to

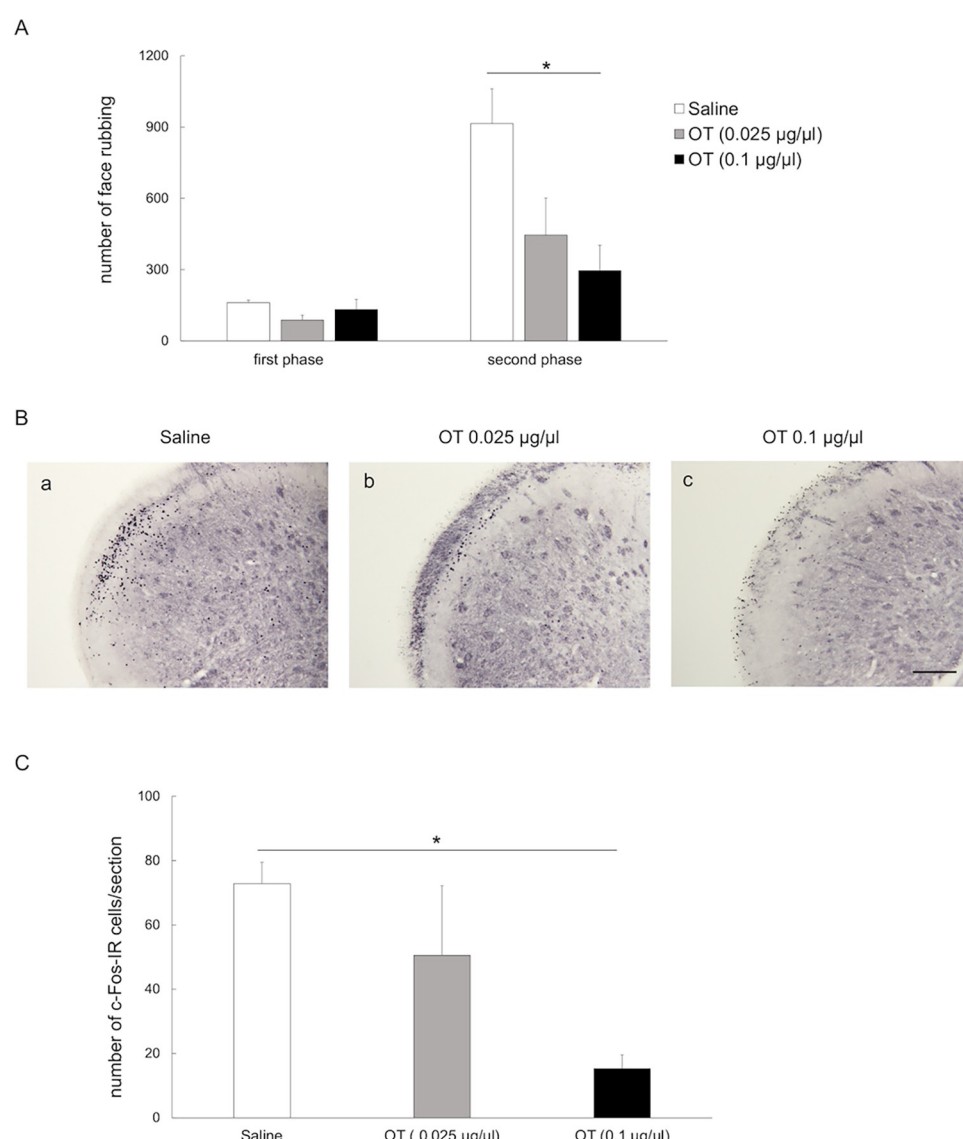

**Fig 8. Effect of intracisternal administration of OT on frequency of face-rubbing and number of c-Fos-IR cells after orofacial formalin test.** (A) Intracisternal administration of OT (0.025 or 0.1 µg/µl) did not affect the frequency of face-rubbing in the first phase in 6-OHDA-lesioned rats in the formalin group. However, the frequency of face-rubbing was significantly decreased in the second phase in 6-OHDA-lesioned rats receiving 0.1 µg/µl OT compared to those receiving saline. First phase: 10–45 minutes; second phase: 10–45 minutes after SC injection of formalin into the left vibrissa pad; OT: oxytocin; c-Fos-IR: c-Fos immunoreactive. *p-value < 0.05, one-way ANOVA followed by Bonferroni's test for multiple comparisons. (B) c-Fos immunostaining (using DAB) of the trigeminal spinal subnucleus caudalis (Vc) of 6-OHDA-lesioned rats in the formalin injection group receiving intracisternal saline(a) or OT (b: 0.025 µg/µl, c: 0.1 µg/µl). Photomicrographs show sections from an area 1440 µm caudal to the obex on the ipsilateral side (to 6-OHDA or saline injection). Scale bars: 200 µm. (C) In comparison to saline, intracisternal administration of 0.1 µg/µl OT significantly reduced the number of c-Fos-IR cells in the Vc of 6-OHDA-lesioned rats, while no such effects were observed with 0.025 µg/µl OT. OT: oxytocin. * p-value < 0.05, one-way ANOVA followed by Bonferroni's test for multiple comparisons.

sham-operated rats in the formalin injection group. Comparison of the formalin injection and no injection groups showed a marked increase in the plasma concentration of OT in sham-operated rats but not in 6-OHDA-lesioned rats, while 6-OHDA-lesioned rats in the formalin injection group exhibited significantly lower plasma OT concentration than sham-operated

rats in the same group. These findings suggest attenuation of the activation of OT neurosecretory cells in the ipsilateral mPVN and dpPVN of 6-OHDA-lesioned rats. This, in turn, decreased OT secretion from the PP into the bloodstream, leading to suppression of its analgesic effects and, consequently, hyperalgesia.

OT neurosecretory cells, which are abundant in the PVN, can be activated by nociceptive stimuli, resulting in secretion of OT through either humoral or nerve regulation [33, 34, 63–67]. During humoral regulation, OT from the magnocellular neurosecretory neurons is conveyed to the PP, secreted into the systemic circulation, and then delivered to the target organs [33, 66]. Studies have shown that OT neurosecretory cells receive excitatory synaptic inputs from A1 noradrenergic neurons in the medulla oblongata, and noxious stimuli can facilitate OT release from PP via these inputs [65, 68]. This is evidenced by an increase in the plasma concentration of OT after SC formalin injection into the hind paws of rats [33]. During nerve regulation, OT from the parvocellular neurosecretory neurons in the PVN directly project to other brain regions including the spinal dorsal horn and Vc [22, 33, 69].

OT has two analgesic mechanisms, including the hypothalamic-spinal oxytocinergic pathway and the indirect endogenous opioid system [70]. In the hypothalamic-spinal oxytocinergic pathway, stimulation of the PVN or administration of OT activates presynaptic OT receptors specifically located in the spinal dorsal horn which, in turn, results in excitation of inhibitory GABAergic interneurons and inhibition of sensory glutamatergic transmission and antinociception [33, 66, 69, 71, 72]. OT receptors are also expressed in the dorsal root ganglion, the trigeminal ganglion (TG), and the Vc [66, 72, 73]. Electrophysiological studies have shown that OT inhibits peripherally evoked neuronal activity in the Vc through activation of OT receptors [73]. In the indirect endogenous opioid system, OT is involved in the supraspinal modulation of inflammatory pain through μ- and κ-opioid receptors [74].

Several studies have investigated the analgesic effects of OT in humans and animals, with some human studies reporting analgesic effects associated with OT and others observing the opposite [75–83]. Animal studies, however, have suggested that intrathecal, intraventricular fourth, intranasal, subcutaneous and trigeminal administration of OT produces an analgesic effect [56, 66, 72, 74, 80, 82, 84, 85]. A previous study examining rats with inflammation induced by SC injection of carrageenan into the hind paw showed that intrathecal administration of OT had an analgesic effect in response to thermal and mechanical stimuli, and this effect peaked at 5 minutes and persisted for 1 hour [56]. Another study investigating rats with inflammation induced by SC injection of formalin into the vibrissa pad 4 minutes after OT administration into the 4th ventricle showed reduction of face-rubbing in the first and second phases [74]. To the best of our knowledge, the current study is the first to show that intracisternal administration of OT had a dose-dependent analgesic effect after SC injection of formalin, as evidenced by a decrease in the number of c-Fos-IR cells in the Vc and the frequency of face-rubbing in the second phase but not in the first phase. In contrast to previous evidence, analgesic effects were only observed in the second phase in the current study and this could be attributed to differences in study methodologies including the method of administration of OT and the concentration of formalin solution used. The first phase is thought to occur due to direct chemical activation of nociceptive afferent fibers, while the second phase is a consequence of activation of central sensitized neurons by peripheral inflammation and ongoing activity of the primary afferents [86]. The endogenous μ-opioid receptor system is activated by formalin injection and produces an anti-nociceptive effect only in the second phase [87]. OT is involved in the supraspinal modulation of inflammatory pain through endogenous μ-opioid receptors [74]. These findings can potentially explain why OT had an analgesic effect only in the second phase of the formalin test in the current study. Nevertheless, intracisternal administration of OT was seen to relieve hyperalgesia in PD rats. Although this finding is useful, there are

limitations in proving that the cause of hyperalgesia in PD rats is the attenuation of OT neuro-secretory cell activation. An electrophysiological approach, for example, direct electrical stimu-lation of the PVN in PD rats, may be necessary.

OT receptors are expressed in various regions of the brain, including TG and Vc neurons in the trigeminal system [88, 89]. Microinjection of OT receptor antagonists into the caudate nucleus can decrease the pain threshold [90], while microinjection of OT into the anterior cin-gulate cortex can attenuate nociceptive and anxiety-like responses in animals with peripheral nerve injury [91]. Microinjection of OT into the TG can attenuate mechanical hypersensitivity [88]. There is limited evidence of the relationship between PD and OT receptors. In the cur-rent study, intracisternal OT administration may have resulted in the activation of OT recep-tors expressed in multiple regions of the brain instead of just one. OT receptors have a high degree of sequence homology to vasopressin-1A (V1A) receptors [92]. A previous study dem-onstrated the alleviation of mechanical hypersensitivity induced by infraorbital nerve injury upon administration of OT into the TG, and this was mediated by V1A receptors in the TG [72]. In the current study, OT may have affected hyperalgesia in PD model rats through the V1A receptors.

PD patients can exhibit neurodegeneration of dopaminergic nerves, and previous studies have demonstrated an association between dopamine and OT, both of which are known to have analgesic effects [3, 4, 6, 7, 12, 93]. Evidence suggests that dopamine agonists can promote peripheral and central OT release, intraventricular injection of dopamine can increase the elec-trical activity of the oxytocinergic neurons, and PVN possesses dopamine D2 receptors [94, 95]. The analgesic effects of dopamine can be attributed to modulation of trigeminal neuro-pathic pain by the descending dopaminergic control system [96]. The number of OT-immu-noreactive neurons in the PVN of PD patients has also been shown to be lower than that of control patients, suggesting an association between PD and OT [97].

## Conclusions

The findings of the current study show that hyperalgesia in PD rats can be attributed to sup-pression of the analgesic effects of OT originating from the PVN, and OT administration can attenuate this.

## Supporting information

**S1 File. Raw data of behavioral and immunohistochemical responses.**
(XLSX)

## Acknowledgments

The authors thank Lecturer Ayano Katagiri (Department of Oral Physiology, Osaka University Graduate School of Dentistry), and Emeritus Professor Atsushi Yoshida (Department of Sys-tematic Anatomy and Neurobiology, Osaka University Graduate School of Dentistry and Department of Oral Health Sciences, Faculty of Health Care Sciences, Takarazuka University of Medical and Health Care), for their constructive suggestions, technical support, and encour-agement. The authors also grateful to Enago (https://www.enago.jp) for English language editing.

## Author Contributions

**Conceptualization:** Nayuka Usami, Hiroharu Maegawa, Hitoshi Niwa.

**Data curation:** Nayuka Usami, Hiroharu Maegawa, Masayoshi Hayashi, Hitoshi Niwa.

**Formal analysis:** Nayuka Usami, Hiroharu Maegawa, Hitoshi Niwa.

**Funding acquisition:** Hitoshi Niwa.

**Investigation:** Nayuka Usami, Hiroharu Maegawa, Masayoshi Hayashi.

**Methodology:** Nayuka Usami, Hiroharu Maegawa, Chiho Kudo.

**Project administration:** Nayuka Usami, Hiroharu Maegawa, Chiho Kudo.

**Resources:** Nayuka Usami, Hiroharu Maegawa.

**Software:** Nayuka Usami.

**Supervision:** Nayuka Usami, Hiroharu Maegawa, Hitoshi Niwa.

**Validation:** Nayuka Usami, Hiroharu Maegawa, Hitoshi Niwa.

**Visualization:** Nayuka Usami, Hiroharu Maegawa, Hitoshi Niwa.

**Writing – original draft:** Nayuka Usami, Hiroharu Maegawa, Hitoshi Niwa.

**Writing – review & editing:** Nayuka Usami, Hiroharu Maegawa, Hitoshi Niwa.

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
