## [Decision Letter · Decision Letter 0]

10 Apr 2024

PONE-D-24-06953Changes in the analgesic mechanism of oxytocin can contribute to hyperalgesia in Parkinson’s disease model ratsPLOS ONE

Dear Dr. Usami,

Thank you for submitting your manuscript to PLOS ONE. After careful consideration, we feel that it has merit but does not fully meet PLOS ONE’s publication criteria as it currently stands. Therefore, we invite you to submit a revised version of the manuscript that addresses the points raised during the review process.

We look forward to receiving your revised manuscript.

Kind regards,

Oscar Arias-Carrion, M.D., Ph.D

Academic Editor

PLOS ONE

“We received funding from JSPS KAKENHI (Grant number: 26463062).”

Reviewers' comments:

Reviewer's Responses to Questions

**Comments to the Author**

1. Is the manuscript technically sound, and do the data support the conclusions?

Reviewer #1: Yes

Reviewer #2: Partly

2. Has the statistical analysis been performed appropriately and rigorously? 

Reviewer #1: Yes

Reviewer #2: Yes

3. Have the authors made all data underlying the findings in their manuscript fully available?

Reviewer #1: Yes

Reviewer #2: Yes

4. Is the manuscript presented in an intelligible fashion and written in standard English?

Reviewer #1: Yes

Reviewer #2: Yes

5. Review Comments to the Author

Reviewer #1: The study examines the role of the descending pain inhibitory system (DPIS) and the involvement of oxytocin (OT) in mitigating hyperalgesia in a rat model of Parkinson's disease (PD) with dopaminergic nigrostriatal lesions induced by 6-OHDA. The findings suggest that PD may lead to hyperalgesia through the suppression of the analgesic effects of OT originating from the paraventricular nucleus (PVN).

Overall, I find that the manuscript is well-written, although I believe it has some minor issues that need to be addressed:

1. While the article's purpose becomes evident upon thorough reading, the initial introduction fails to fully elucidate the research objective and hypothesis. It is suggested that the final paragraph of the introduction be revised to ensure clarity regarding these aspects.

2. In both the abstract and conclusion of the manuscript, it is stated that "these findings confirm the presence of hyperalgesia in PD patients." However, it is important to note that the experiments were conducted solely on rats, and therefore, extrapolation to human beings should be approached cautiously. I would recommend modifying the phrases to specify that the findings are limited to the rat model.

3. The methods section lacks clarity regarding the total number of animals used and their respective groupings, which is crucial for ensuring reproducibility. Providing a detailed description, possibly aided by a diagram, would enhance clarity in this regard. Additionally, it's important to address why male rats were exclusively chosen for the experiments and why female animals were not included in the study. Clarifying these points would strengthen the experimental design and broaden the applicability of the findings.

4. TH immunohistochemical staining is not described in the methods.

5. The Statistical Analyses section could benefit from further elaboration to specify the precise points where Welch’s t-test, one-way ANOVA, and two-way ANOVA tests were employed. Alternatively, this information could be integrated into the Results section to enhance clarity regarding the statistical methods utilized and their respective applications.

Reviewer #2: Review comments:

In this manuscript the authors examined the role of oxytocin (OT) in the orofacial hyperalgesia induced by the injection of 6-OHDA in male rats. The paper is thematic to the journal. There are number of issues that could be addressed to improve the work.

Line120. Three weeks after 6-OHDA or saline injection into the left MFB, an orofacial formalin test was performed to allow measurement of the response to chemical stimulation.

It has been reported that three weeks after administering 6-OHDA, rats present hyperalgesia and mechanical allodynia (Romero Sánchez et al., 2020). What is the time course of orofacial hyperalgesia after administering 6-OHDA?

Line 275. The mean frequency of face-rubbing in the first and second phases after SC injection of formalin or saline into the left vibrissa pad was recorded.

Which is the temporal course of the development of hyperalgesia induced by formalin?

Line 276. No significant difference in the frequency of face-rubbing was observed between the 6-OHDA-lesioned and sham-operated rats in the saline injection group..

How do you explain that there is not a significant difference in the frequency of the face-rubbing between the 6-OHDA-lesiones and sham operated rats after 3 weeks of 6-OHDA injection?

Could the no difference in the frequency of face-rubbing be due to the method used to measure nociceptive behavior?

It has been reported that in the PD model, orofacial mechanical allodynia occurs 4 days after applying 6-OHDA (Dieb et al., 2015, 2018). Also, the administration of 6-OHDA produced mechanical allodynia in the two hind limbs from day 18 and mechanical hyperalgesia from 3 days after the rats were injected (Romero Sánchez et al., 2020).

Are the cellular mechanisms underlying allodynia different from those that produce hyperalgesia?

Line 279. In contrast, 6-OHDA-lesioned rats exhibited significantly higher frequency of face-rubbing than the sham-operated rats in the formalin injection group.

How long did formalin-induced hyperalgesia persist?

Line 285. Comparison of the 6-OHDA-lesioned and sham-operated rats showed no significant differences in the number of c-Fos-IR cells in the Vc in the no injection and saline injection groups

Does this mean that after three weeks of 6-OHDA administration central sensitization has still not occurred?

How could the development of allodynia reported by Dieb et al. (2015) be explained in this PA model if there are no significant differences in the number of c-Fos-IR cells in the Vc between the 6-OHDA-lesioned and sham-operated rats?

Line 317. The number of c-Fos-IR cells in the DPIS nuclei (i.e., vlPAG, LC, NRM, and PVN) of the 6-OHDA-lesioned and sham-operated rats receiving formalin injections were counted (Fig 3A), and no significant differences were observed except in the PVN where a significant decrease in the number of c-Fos-IR cells was observed on the ipsilateral side of the 6-OHDA-lesioned rats

Before formalin application, how is the expression of c-Fos in PVN in sham rats and in those injured with 6-OHDA?

Line 350. injection groups (Fig 4A). In the formalin injection group, the 6-OHDA-lesioned rats exhibited significant decreases in the number of c-Fos-IR cells in the ipsilateral mPVN when compared to the sham-operated rats

After formalin administration, for how long is the decrease in the number of c-Fos-IR cells maintained in the ipsilateral mPVN of rats administered with 6-OHDA?

Line 394. in the formalin injection group, significant decreases in the percentage of OT-IR neurons co-localized with c-Fos protein were observed in the ipsilateral mPVN and dpPVN of 6-OHDA-lesioned rats compared to sham-operated rats…

After formalin administration, for how long is the decrease in the percentage of OT-IR neurons colocalized with c-Fos-IR cells maintained in the ipsilateral mPVN and dpPVN of rats administered with 6-OHDA?

Line 503. Comparison of the formalin injection and no injection groups showed a marked increase in the plasma concentration of OT in sham-operated rats but not in 6-OHDA-lesioned rats, while 6-OHDA-lesioned rats in the formalin injection group exhibited significantly lower plasma OT concentration than sham-operated rats in the same group. These findings suggest attenuation of the activation of OT neurosecretory cells in the ipsilateral mPVN and dpPVN of 6-OHDA-lesioned rats. This, in turn, decreased OT secretion from the PP into the bloodstream, leading to suppression of its analgesic effects and, consequently, hyperalgesia.

What does it mean that the plasma concentration of OT in the no injection group of 6-OHDA lesioned rats seems to be higher or equal to that of the sham-operated rats?

6. PLOS authors have the option to publish the peer review history of their article (what does this mean?). If published, this will include your full peer review and any attached files.

Reviewer #1: **Yes: **Emmanuel Ortega-Robles

Reviewer #2: No

---

## [Author Response · Author response to Decision Letter 0]

15 Jun 2024

Comment 1: Please ensure that your manuscript meets PLOS ONE's style requirements, including those for file naming. 

Response: Thank you for this pertinent remark. The name of the Figure has been revised (“Fig 1.tiff” → “Fig1.tif”). Moreover, the formula on line 245 has been changed. The name of the “S1 file.xlsx” was also changed to “S1_File.xlsx.” Supporting information has also been moved to the last page. The funding or competing interests statement have also been removed. We have endeavored to follow the guidelines of PLOS ONE submission; however, we remain available to address any concerns you have regarding the formatting of our manuscript.

Comment 2: “We received funding from JSPS KAKENHI (Grant number: 26463062).”

Please state what role the funders took in the study. 

Response: Thank you for this insightful comment. We apologize for not stating the role of the funders. They were involved in Conceptualization, Data curation, Formal analysis, Supervision, Validation, Visualization, and Writing – review & editing.

Comment 3: When completing the data availability statement of the submission form, you indicated that you will make your data available on acceptance. We strongly recommend all authors decide on a data sharing plan before acceptance, as the process can be lengthy and hold up publication timelines. Please note that, though access restrictions are acceptable now, your entire data will need to be made freely accessible if your manuscript is accepted for publication. This policy applies to all data except where public deposition would breach compliance with the protocol approved by your research ethics board. If you are unable to adhere to our open data policy, please kindly revise your statement to explain your reasoning and we will seek the editor's input on an exemption. Please be assured that, once you have provided your new statement, the assessment of your exemption will not hold up the peer review process.

Response: Thank you for this pertinent remark. We understand.

 

Reviewers” comments:

Reviewer #1

Comment 1: While the article's purpose becomes evident upon thorough reading, the initial introduction fails to fully elucidate the research objective and hypothesis. It is suggested that the final paragraph of the introduction be revised to ensure clarity regarding these aspects.

Response: Thank you for your comments. The final section of the introduction has been revised to ensure clarity regarding the study objectives and hypotheses. Additionally, this study focuses on oxytocin in the PVN, and to focus the perspective, the description of corticotropin-releasing hormone in lines 73–78 from the third paragraph of the Introduction has been deleted.

Comment 2: In both the abstract and conclusion of the manuscript, it is stated that "these findings confirm the presence of hyperalgesia in PD patients." However, it is important to note that the experiments were conducted solely on rats, and therefore, extrapolation to human beings should be approached cautiously. I would recommend modifying the phrases to specify that the findings are limited to the rat model.

Response: We have revised “patients” to “rats” in line 38 of the abstract and in line 658 of the conclusion.

Comment 3: The methods section lacks clarity regarding the total number of animals used and their respective groupings, which is crucial for ensuring reproducibility. Providing a detailed description, possibly aided by a diagram, would enhance clarity in this regard. Additionally, it's important to address why male rats were exclusively chosen for the experiments and why female animals were not included in the study. Clarifying these points would strengthen the experimental design and broaden the applicability of the findings.

Response: Thank you for your pertinent remarks. A new Figure 1 has been created, specifying the number of rats used for each experiment. Furthermore, a new section on the study protocol has been added to line 140 of the methods section to make the experimental flow easier to understand. 

Only male rats were used in the present study because they were the only ones used in the previous study and it was necessary to conduct the experiment under the same conditions as it was conducted previously. It will certainly be necessary in the future to investigate whether the same results can be achieved in a similar study on hyperalgesia involving female rats as it was in male rats (ine 105-110).

Comment 4: TH immunohistochemical staining is not described in the methods.

Response: Thank you for the suggestion, but we have described the TH staining method in lines 195–207.

Comment 5: The statistical analyses section could benefit from further elaboration to specify the precise points where Welch’s t-test, one-way ANOVA, and two-way ANOVA tests were employed. Alternatively, this information could be integrated into the Results section to enhance clarity regarding the statistical methods utilized and their respective applications.

Response: Thank you for your remark. Statistical methods are described in more detail in the Methods section. They are also newly described in the description section of each figure.

 

Reviewers” comments:

Reviewer #2: 

Comment 1: Line120. Three weeks after 6-OHDA or saline injection into the left MFB, an orofacial formalin test was performed to allow measurement of the response to chemical stimulation.

It has been reported that three weeks after administering 6-OHDA, rats present hyperalgesia and mechanical allodynia (Romero Sánchez et al., 2020). What is the time course of orofacial hyperalgesia after administering 6-OHDA?

Response: Thank you for your comments. With regard to the time course of facial hyperalgesia after 6-OHDA administration, we unfortunately could not find any reports investigating the time course of the formalin test. However, we found several papers that examined the time course after 6-OHDA administration for other stimuli.

According to Takeda, in the side ipsilateral to 6-OHDA injections, unilateral PD model rats showed a shorter latency of the withdrawal response to the mechanical stimulus to the plantar surface of the hindpaw at one, four, and twelve weeks after 6-OHDA lesioning compared with that of sham-operated rats [1]. According to Domenich et al, in the ipsilateral side to 6-OHDA injections, unilateral PD model rats showed a mechanical hyperalgesia (measured by paw pressure test) to the hind paw at 7, 14, 21 days after 6-OHDA administration compared with control rats [2]. As regards the orofacial hyperalgesia, according to Dieb et al, in the ipsilateral side to 6-OHDA injections, unilateral PD model rats showed a mechanical allodynia to the vibrissa pad at 4 days to 6 weeks after 6-OHDA lesioning compared with that of sham rats.

Most of the papers that have conducted formalin tests using PD models have done so using rats three to six weeks after administration of 6-OHDA [4,5,6]. In the previous paper of this study, a formalin test was performed 3 weeks after 6-OHDA administration, so it was necessary to time the formalin test to investigate the mechanism of hyperalgesia in PD rats under the same conditions. [4]. Considering these papers, we consider that the formalin test performed in the present study three weeks after 6-OHDA administration is an appropriate time for stimulation.

[1] Takeda et al. Unilateral lesions of mesostriatal dopaminergic pathway alters the withdrawal response of the rat hindpaw to mechanical stimulation. Neuroscience Research. 2005; 52: 31-36.

[2] Domenici et al. Parkinson’s disease and pain: Modulation of nociceptive circuitry in a rat model of nigrostriatal lesion. Experimental Neurology. 2019; 315: 72-81.

[3] Dieb et al. Nigrostriatal dopaminergic depletion produces orofacial static mechanical allodynia. European journal of pain. 2016; 30: 196-205. 

[4] Maegawa et al. Neural mechanism underlying hyperalgesic response to orofacial pain in Parkinson's disease model rats. Neurosci Res. 2015;96: 59-68.

[5] Chudler et al. Nociceptive behavioral responses to chemical, thermal and mechanical stimulation after unilateral, intrastriatal administration of 6-hydroxydopamine. Brain research. 2008; 1213: 41-47.

[6] Tassorelli et al. Most of the papers that have conducted formalin tests using PD models have done so using rats three to six weeks after administration of 6-OHDA. Brain research. 2007; 1176: 53-61.

Comment 2: Line 275. The mean frequency of face-rubbing in the first and second phases after SC injection of formalin or saline into the left vibrissa pad was recorded.

Which is the temporal course of the development of hyperalgesia induced by formalin?

Response: Thank you for your remark. We have newly created a time-course graph of face rubbings after SC injection of formalin or saline, in Figure 3A. More detailed data can be found in “S1 _File.xlsx” of the supporting information.

Comment 3: Line 276. No significant difference in the frequency of face-rubbing was observed between the 6-OHDA-lesioned and sham-operated rats in the saline injection group.

How do you explain that there is not a significant difference in the frequency of the face-rubbing between the 6-OHDA-lesiones and sham operated rats after 3 weeks of 6-OHDA injection?

Could the no difference in the frequency of face-rubbing be due to the method used to measure nociceptive behavior?

Response: Thank you for the suggestion. The saline SC injection group in this study was created as a control group for the formalin SC injection group. The reason for creating a control group, it has been reported that as formalin-induced face rubbing is sometimes bilateral and resemble normal face washing, a control group without formalin injection (i.e. saline injection group) should be used to quantify spontaneous, symmetrical face washing and provide a 'pain-free' baseline when performing orofacial formalin test [1, 2] (line 334-340). Furthermore, we considered it necessary to first examine whether the hyperalgesia induced by formalin in 6OHDA rats was really due to the chemical stimulation of formalin. In other words, we thought it necessary to prove that tissue damage caused purely by the chemical stimulation of formalin causes hyperalgesia in 6-OHDA-lesioned rats, eliminating the effects of 'needle penetration into the skin' and 'volume by liquid injection into the subcutaneous mucosa' during formalin SC injection. All of this suggests that the fact that the 6-OHDA-lesioned rats are hyperalgesic after formalin injection but no significant difference between them after saline injection when compared between 6OHDA-lesioned and sham-operated rats is not a problem at all, but rather proof that hyperalgesia is caused by the chemical stimulation of formalin.

[1] Chudler EH, Lu Y. Nociceptive behavioral responses to chemical, thermal and mechanical stimulation after unilateral, intrastriatal administration of 6-hydroxydopamine. Brain Res. 2008;1213: 41-47.

[2] Raboisson P and Dallel R. The orofacial formalin test. Neurosci Biobehav Rev. 2004; 28: 219–226.

Comment 4: It has been reported that in the PD model, orofacial mechanical allodynia occurs 4 days after applying 6-OHDA (Dieb et al., 2015, 2018). Also, the administration of 6-OHDA produced mechanical allodynia in the two hind limbs from day 18 and mechanical hyperalgesia from 3 days after the rats were injected (Romero Sánchez et al., 2020).

Are the cellular mechanisms underlying allodynia different from those that produce hyperalgesia?

Response: Thank you for your remark. Hyperalgesia and allodynia are defined differently. The hyperalgesia according to the formalin test means that 'everyone feels pain as a result of painful stimuli, but in hyperalgesia the pain is felt more intensely'. On the other hand, allodynia is defined as pain produced by a normally non-noxious stimulus, can be triggered by routine activities such as shaving, teeth cleaning, chewing, talking, or even exposure to air currents, and it considerably reduces the patient’s quality of life [1]. 

In terms of differences in mechanisms, formalin is a chemical stimulus, activates Aδ and C nociceptors as well as spinal and trigeminal nociceptive neurons and provokes concentration-dependent tissue damage [2]. Injected in the upper lip of the rat, it generates characteristic behavioral responses consisting of recurrent and persistent episodes of paw strokes directed to the perinasal area (face rubbing)[2]. On the other hand, Aδ-afferent neurons in trigeminal ganglion are significantly involved in the changes in afferent spontaneous activity and mechanically evoked activity that accompany mechanical allodynia [1].

[1] Dieb et al. PKCγ-Positive Neurons Gate Light Tactile Inputs to Pain Pathway Through pERK1/2 Neuronal Network in Trigeminal Neuropathic Pain Model. J Oral Facial Pain Headache. 2015;29:70-82.

[2] Raboisson P and Dallel R. The orofacial formalin test. Neurosci Biobehav Rev. 2004; 28: 219–226.

Comment 5: Line 279. In contrast, 6-OHDA-lesioned rats exhibited significantly higher frequency of face-rubbing than the sham-operated rats in the formalin injection group.

How long did formalin-induced hyperalgesia persist?

Response: Thank you for your remark. We have newly created a time-course graph of face rubbings after SC injection of formalin or saline, in Figure 3A. More detailed data can be found in “S1 _File.xlsx” of the supporting information.

Comment 6: Line 285. Comparison of the 6-OHDA-lesioned and sham-operated rats showed no significant differences in the number of c-Fos-IR cells in the Vc in the no injection and saline injection groups

Does this mean that after three weeks of 6-OHDA administration central sensitization has still not occurred?

How could the development of allodynia reported by Dieb et al. (2015) be explained in this PA model if there are no significant differences in the number of c-Fos-IR cells in the Vc between the 6-OHDA-lesioned and sham-operated rats?

Response: Thank you for your comments. The response to this comment is almost similar to the response to comment 3. Allodynia is a painful response to a normally non-painful stimulus and is described as a lowering of the threshold. In this study, saline was injected subcutaneously into the vibrissa pad, and although pain was felt due to the needle puncture during subcutaneous injection, no inflammatory reaction or tissue destruction occurred as the content was saline and not an irritant, and this did not manifest as pain behavior. Comparing 6-OHDA-lesioned rats with sham-operated rats, the fact that 6-OHDA-lesioned rats show hyperalgesia after formalin injection but no significant change after saline injection is not a problem at all, but rather proof that hyperalgesia is caused by the chemical stimulation of formalin. The results of the present study cannot be compared with those of Dieb et al.'s study of allodynia in 6-OHDA rats, as the type of pain stimulus is different.

Comment 7: Line 317. The number of c-Fos-IR cells in the DPIS nuclei (i.e., vlPAG, LC, NRM, and PVN) of the 6-OHDA-lesioned and sham-operated rats receiving formalin injections were counted (Fig 3A), and no significant differences were observed except in the PVN where a significant decrease in the number of c-Fos-IR cells was observed on the ipsilateral side of the 6-OHDA-lesioned rats

Before formalin application, how is the expression of c-Fos in PVN in sham rats and in those injured with 6-OHDA?

Response: Thank you for your pertinent remarks. The number of c-Fos-immunoreactive cells/ section in the PVN in the no injection group of the 6-OHDA-lesioned rats and sham-operated rats was counted and a new graph was created as Fig. 4C (line 398-410 ).

Comment 8: Line 350. injection groups (Fig 4A). In the formalin injection group, the 6-OHDA-lesioned rats exhibited significant decreases in the number of c-Fos-IR cells in the ipsilateral mPVN when compared to the sham-operated rats

After formalin administration, for how long is the decrease in the number of c-Fos-IR cells maintained in the ipsilateral mPVN of rats administered with 6-OHDA?

Response: Thank you for your remark. Matsuura et al. reported that subcutaneous inj

---

## [Editor Report · Decision Letter 1]

20 Jun 2024

Changes in the analgesic mechanism of oxytocin can contribute to hyperalgesia in Parkinson’s disease model rats

PONE-D-24-06953R1

Dear Dr. Usami,

We’re pleased to inform you that your manuscript has been judged scientifically suitable for publication and will be formally accepted for publication once it meets all outstanding technical requirements.

Kind regards,

Oscar Arias-Carrion, M.D., Ph.D

Academic Editor

PLOS ONE
---

## [Editor Report · Acceptance letter]

10 Aug 2024

PONE-D-24-06953R1 

PLOS ONE

Dear Dr. Usami, 

I'm pleased to inform you that your manuscript has been deemed suitable for publication in PLOS ONE. Congratulations! Your manuscript is now being handed over to our production team.

Kind regards, 

on behalf of

Prof. Dr. Oscar Arias-Carrion 

Academic Editor

PLOS ONE